# Assessing Credibility in Bayesian Networks Structure Learning

**DOI:** 10.3390/e26100829

**Published:** 2024-09-30

**Authors:** Vitor Barth, Fábio Serrão, Carlos Maciel

**Affiliations:** 1Department of Electrical and Computing Engineering, University of Sao Paulo, São Carlos 13566-590, SP, Brazil; 2Department of Physical Therapy, Federal University of São Carlos, São Carlos 13565-905, SP, Brazil; fserrao@ufscar.br; 3Department of Electrical Engineering, State University of São Paulo, Guaratinguetá 12516-410, SP, Brazil; carlos.maciel@unesp.br

**Keywords:** Bayesian networks, explainable models, probabilistic learning

## Abstract

Learning Bayesian networks from data aims to create a Directed Acyclic Graph that encodes significant statistical relationships between variables and their joint probability distributions. However, when using real-world data with limited knowledge of the original dynamical system, it is challenging to determine if the learned DAG accurately reflects the underlying relationships, especially when the data come from multiple independent sources. This paper describes a methodology capable of assessing the credible interval for the existence and direction of each edge within Bayesian networks learned from data, without previous knowledge of the underlying dynamical system. It offers several advantages over classical methods, such as data fusion from multiple sources, identification of latent variables, and extraction of the most prominent edges with their respective credible interval. The method is evaluated using simulated datasets of various sizes and a real use case. Our approach was verified to achieve results comparable to the most recent studies in the field, while providing more information on the model’s credibility.

## 1. Introduction

Bayesian networks (BNs) are graph-based models that use probability theory to represent a collection of variables and their conditional relationships [1]. They provide an efficient way to describe and compute joint probabilities by exploiting conditional independence. With existing and reliable evidence, BNs can generate predictions by calculating the probabilities of different outcomes [1]. They are also able to uncover uncertain relationships based on incomplete or missing data, effectively representing the inherent uncertainty in various real-world problems [2], which can then be enhanced by integrating expert knowledge as an a priori probability of the network structure.

Due to their ability to model complex probabilistic relationships and, in particular, probabilistic reasoning [3], BNs play a significant role in Artificial Intelligence (AI). They provide a structured way of representing and reasoning about uncertainty, making it valuable for applications where interpretability is essential, such as healthcare, finance, and risk management. Bayesian networks are also important for integrating machine learning algorithms with domain knowledge; this integration allows for more accurate and reliable models, as domain experts can guide the learning process and improve the interpretability of the results. For example, combining machine learning with expert knowledge using BNs in risk analysis leads to better risk assessment and decision making [4].

Bayesian parameter estimation methods are often used to learn Bayesian networks. These techniques combine prior knowledge with observed data, enabling the models to adjust and revise their beliefs as they receive new information. The performance and adaptability of BN models are improved through an iterative process of learning and refinement as additional data streams become available, which is particularly important for evolving and adaptive intelligent systems [5]. The extraction of a BN from data can be achieved using different procedures and approaches [6], but, in general, the key objective is to identify the structural model, which is depicted by nodes and edges in a Directed Acyclic Graph (DAG), as well as the set of conditional probabilities called BN parameters [7]. The BN model can handle discrete or continuous distributions [8].

Learning BN structures from data is crucial for various reasons: The structure of a BN can help uncover causal relationships between different variables, providing an understanding of the underlying mechanisms of the modeled system [9]; even though the structural model of a BN does not present a causal graph, it can still serve as a foundation for analysis. The dilemma of data quantity and edge credibility in Bayesian networks arises from balancing two key factors: having sufficient data to estimate the conditional probability distributions accurately and ensuring the network structure (edges) accurately represents the true dependencies between variables. Bayesian networks require significant data to reliably estimate the conditional probability tables (CPTs) or distributions associated with each node. When there are insufficient data, the probability estimates can be imprecise, leading to inaccurate inferences from the network [10,11]. The amount of data needed depends on the number of variables, their cardinalities (number of states), and the complexity of the network structure. More data are required as the network grows more extensive and more complex. However, there is no fixed formula to determine the ideal data quantity, as it depends on the specific problem.

The edges in a Bayesian network represent the conditional dependencies between variables. These dependencies are crucial for the network’s structure and the accuracy of its inferences. Edges can be constructed manually based on expert knowledge or learned from data using structure learning algorithms [10,12]. Manually constructed edges are more credible because they leverage expert domain knowledge. However, this process is time-consuming, subjective, and prone to human errors. Data-driven learning requires substantial amounts of data to ensure that inferred dependencies are accurate. The dilemma arises because ensuring edge credibility often requires a large amount of data, but obtaining such extensive data can be impractical or impossible in many real-world scenarios. Even with abundant data, the computational cost of inference can become prohibitive as the network becomes more complex. Conversely, if the data are limited, the estimated probabilities may be inaccurate and the learned network structure may not accurately represent the proper dependencies, leading to unreliable inferences.

Model structural accuracy is frequently overlooked, as many algorithms, such as PC or Local Search algorithms, are treated as straightforward optimization problems. This approach can lead to models with high levels of uncertainty being treated as accurate representations of the system, given that these algorithms do not provide credibility measures. Furthermore, the possibility of confounding variables may be disregarded since evaluating a single high-scoring model does not account for the probabilities associated with edge directions in all models with similar performance. These issues could be addressed by incorporating Bayesian credible interval analysis during the learning process of Bayesian networks.

This paper introduces a new method to address the absence of a complete Bayesian methodology for learning Bayesian networks. The proposed method employs a Markov chain Monte Carlo (MCMC) technique to assess the learning of BN structures from data. The objective is to offer a Bayesian approach that determines the most critical edges and their orientations while also quantifying the credibility level in each edge’s existence and direction. Using a Dirichlet-Multinomial distribution allows us to evaluate the characteristics of each edge while learning. This helps us to comprehend the various potential arrangements identified by score- or constraint-based techniques. This makes it possible to generate a summary that highlights each connection’s role in the model.

The proposed methodology is based on data-driven learning processes across different subsets of independent and identically distributed (i.i.d.) data, collected from an observed system. This evaluation intends to determine whether the same probabilistic relationships consistently emerge. If these relationships vary, it indicates that no single network can accurately model the system. Consequently, models that display significant structural changes in response to minor data disturbances should be considered less credible than more robust alternatives. Additionally, we suggest that if an edge is highly likely to exist but its direction is uncertain, this may indicate the presence of a latent variable within the dataset. This hypothesis is derived from the analysis of the FCI algorithm [13,14].

The main contributions of this work can be summarized as follows:We introduce a new algorithm that utilises MCMC methods to assess learning about BN structures. Our method overcomes the drawbacks of conventional score-based algorithms, which often do not provide insight into the reasons behind selecting a particular structure over others. Instead, our approach offers a more thorough analysis of possible structures by treating the BN’s structure as changeable and treating the presence or absence of each edge as random variables.We propose the usage of a bootstrapping method for generating multiple subsets of the data, in order to learn a collection of BNs from distinct data samples, which helps capture various aspects of the underlying relationships and enhances the reliability of the learned model.We suggest a Dirichlet-Multinomial model to represent the probability distribution of edge characteristics. This model is based on the observed counts in the learned BNs, providing a robust and flexible framework for quantifying the uncertainty associated with each edge.We demonstrate the effectiveness of our approach on both synthetic and real-life datasets. By comparing the structure of the learned BNs with previously known structures and evaluating the inference capabilities of the final BN, we show that our method achieves competitive results while providing valuable information on the model’s credibility.

The paper is organized as follows: Section 2 provides an overview of Bayesian networks. Section 3 presents the details of our proposed approach, including the algorithmic steps and the MCMC-based edge direction distribution sampling. Section 4 describes the experimental setup and presents the results and discussion. Finally, Section 5 concludes the paper and discusses future research directions.

## 2. Brief Overview on Bayesian Networks

In this section, we will introduce the fundamental concepts of Bayesian networks (BNs), focusing on their structure, mathematical foundations, and how they deal with uncertainty.

### 2.1. Mathematical Framework

The main difference between Bayesian networks and other graphical models is the usage of Directed Acyclic Graphs (DAGs), enforcing the Markov condition, which states that a node only depends on its non-descendants, by preventing the existence of cycles in the BN base graph [15]. This characteristic can be seen as a generalization of a 1st-order Markov chain for DAGs [15,16]; therefore, if a graph meets the Markov condition, each of the nodes is dependent only on their immediate parents, being independent of all its other predecessors [17]. When these conditions are met, it is possible to simplify the model inference, as described below.

Given a DAG G=(V,E) and a Conditional Probability Distribution (CPD) Θ, it is said that a model M(G,Θ) meets the Markov condition if for each variable x∈V, *x* it is conditionally independent ($I_{p}$) of its non-descendants (ND(x)), given the set of its parents (Pa(x)) [1,16],
(1)$$I_{p}\left(x,ND\left(x\right)|Pa\left(x\right)\right)$$
only then can *M* be classified as a Bayesian network.

The usage of DAGs and the observance of the Markov condition, presented in Equation (Equation 1), allows a significant simplification of the JPD calculations, given that the JPDs can be obtained by the product of all the individual functions in Θ [1,16]:(2)$$p\left(x_{1},x_{2},\ldots ,x_{n}\right)=\prod_{k=1}^{n}p\left(x_{k}|Pa\left(x_{k}\right)\right)$$

### 2.2. Learning a Bayesian Network from Data

There are two primary methods for learning structures from data: scoring-based and constraint-based strategies [18]. Score-based methods provide a benchmark to assess adherence between a BN and the modeled data. Afterwards, a search over the domain of DAGs (which cannot be thoroughly explored because of its super-exponential computational complexity ($2^{n^{O(n)}}$) according to the number of variables (n) [18]) is conducted to identify a configuration that attains the highest score. In contrast, constraint-based techniques analyze conditional independence in the data and construct the network accordingly. The learning process is always subject to the quality, granularity, and length of the data, even when estimating models with only discrete variables [19].

Some mixed approaches [20,21,22] to learn the structure of a BN are also possible, such as combining Bayesian networks learned from expert knowledge and data-driven approaches [23]. This can be useful when there are different sources of information or different types of data available. In such cases, exploring the space of plausible structures and measuring the uncertainty associated with each potential structure becomes essential [24]. Methods have already been introduced to quantify the credibility of learned Bayesian networks, as demonstrated in [20]. In this approach, an ensemble of Bayesian networks is learned from bootstrap replicas of the dataset, and an analytical threshold is employed to select the most frequent associations. While being a step in the right direction, it is worth noting that this method still requires many samples, and the depth of information about the credibility of the final model remains limited, since it is not yet a fully Bayesian approach.

The reliance on data quality and length often makes managing small sample sizes a common challenge in Bayesian networks learning [25]. This issue is particularly serious because the most widely used approaches for BN structure learning rely on frequentist statistical techniques, including the PC algorithm, Local Search algorithms, such as hill climbing, and node-ordering strategies. These techniques are well established and effective when data are abundant, but face limitations when used on smaller sample sizes. As a result, there is a demand for new approaches to learning BN from small datasets, or at least methods that offer insights into the credibility of the resulting models. Table 1 outlines the limitations of these algorithms with small sample sizes.

Given the inherent complexity of real-world data, it is important to recognize that a single perfectly accurate model of the observed system may not exist. In fact, when dealing with small datasets, it is often more practical to aim for simpler models that align with the limited evidence available. Attempting to explore the entire space of possible DAGs is not only computationally prohibitive [26], but can lead to overfitting or erroneous conclusions. Instead, focusing on multiple plausible structures that fit the data reasonably well can provide a more nuanced understanding of the underlying relationships. The approach presented by this paper allows for the exploration of alternative models, offering valuable insights into the system’s dynamics, and addressing the uncertainty inherent in learning from limited data [24].

### 2.3. Bayesian Network Uncertainty Metrics

Bayesian network models have a significant role in decision-making, and incorrect structures or inferences can lead to flawed conclusions, which pose great risk in critical areas like medical diagnostics, financial risk assessment, and autonomous systems. Ensuring the credibility of these models is especially necessary in data-driven learning, where models are built on uncertain data [27].

Multiple methodologies exist to compare uncertainty in Bayesian networks, but they primarily aim to evaluate the quality of the inference. Metrics such as confusion error, area under the curve (AUC), and k-fold cross-validation are designed to measure how well a model’s predictions are based on validation datasets [27,28].

Methodologies that focus on model sensitivity and complexity, such as variance reduction and the number of conditional probabilities, indirectly support the learning process by providing information into how model structure and variable interactions affect the inference [27]. Understanding these metrics can guide the selection appropriate variables, setting prior probabilities, and identifying potential sources of uncertainty.

## 3. Materials and Methods

Given the limitations of traditional credibility evaluation metrics, our approach emphasizes the evaluation of edge credibility during the learning process of Bayesian network models. By evaluating the existence and direction of edges, we can better capture the true relationships within the data, resulting in more accurate predictions and a deeper understanding of the underlying system dynamics. This metric also offers information on data quality and quantity, as a larger credible interval may be improved by increasing the amount of data used to inform the model.

The algorithm for calculating this metric is based on sampling a Dirichlet-Multinomial distribution using Markov chain Monte Carlo to uncover the distribution and credibility of each edge in a Bayesian network. The basic steps of the algorithm for a dataset D with samples of *n* random variables $X_{1},X_{2},\ldots ,X_{n}$, are described as follows:Resample D in smaller subsets $d_{1},d_{2},\ldots ,d_{k}$ with independent and identically distributed samples;Learn a set of Bayesian networks $B=\{b_{1},b_{2},\ldots ,b_{k}\}$ using the data for each of the data subsets;Use MCMC to approximate a multinomial distribution for each pair of variables $\left(X_{a},X_{b}\right)$, with the count of occurrence of the edges $\left(X_{a},X_{b}\right)$ or $\left(X_{b},X_{a}\right)$ in *B*.

### Proposed Approach

Learning Bayesian networks structures is usually seen as an optimization problem, and the search space is the set of all possible DAGs formed by the variables. To evaluate the distribution of the edges, it is necessary to have a set of networks to observe. For such a purpose, multiple techniques can be used, whereas a simple but effective one is bootstrapping:

Using Algorithm 1, the original dataset is resampled into multiple smaller sets; for each, a Bayesian network $b_{k}$ is learned. To guarantee an asymptotic behavior, *B* must be complete and, therefore, contain all the possible networks that can explain the data sampled from the observed system.

With the set of BNs *B* ready, the next step is to format the inputs needed for the MCMC sampler. The proposed algorithm is based on learning the distribution of an edge existence and direction across all BNs; therefore, the next step is mapping occurrences of each edge in all of *B*.
**Algorithm 1:** Generating a set *B* containing *k* Bayesian networks from a dataset D, which are learned from *n* samples each. In a stationary and complete dataset, using an ideal scoring method, all networks would be identical. However, real-world data often deviate from this ideal. This algorithm captures small statistical differences within the dataset that may be missed when evaluating it as a whole.
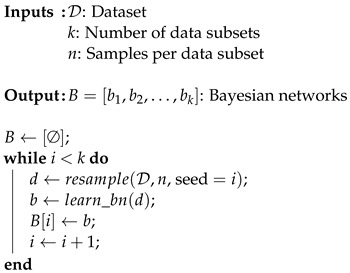


For each possible undirected edge $\left(X_{1},X_{2}\right)$—calculated as the 2 by 2 combination of all random variables—Algorithm 2 outputs an array with three possible states, used to identify the observations:State 1, a left-direction edge: $X_{1}\leftarrow X_{2}$State 2, no edge: $X_{1}≁X_{2}$State 3, a right-direction edge: $X_{1}\to X_{2}$
**Algorithm 2:** Counting the frequency of each edge appearance in *B* (E). This result can later be used as a source in a count data model, so the frequency of each edge is independently analyzed.
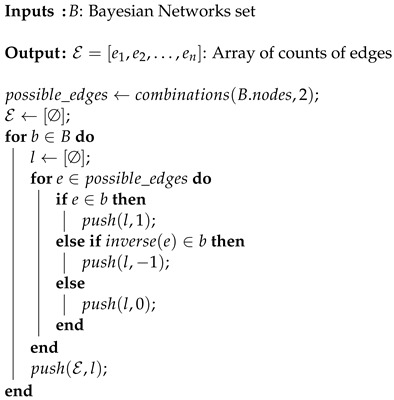


Lastly, MCMC can be used to unravel the distribution of each of the edges evaluated in Algorithm 2:

For each pair of variables $\left(X_{1},X_{2}\right)$, the Algorithm 3 learns a multinomial distribution $\theta_{\left(X_{1},X_{2}\right)}$, containing the probability for each of the three states described above. The Dirichlet distribution is a conjugated prior for the multinomial distribution, since the parameters are unknown; Dirichlet(α=[1/3,1/3,1/3]) is used as an uninformative prior.
**Algorithm 3:** Sampling the distribution of each edge appearance. In this example, a 3-event Dirichlet-Multinomial model is built from an uninformative prior using MCMC.
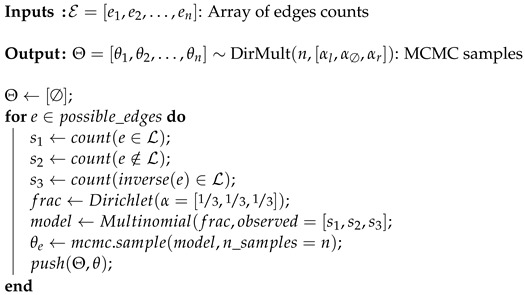


With the distributions of all the edges stored in Θ, each of those can be individually analyzed for the intensity of the relationships and a graph can be constructed with the most probable state of each edge. Note that the result is not necessarily a DAG, since no restrictions on cycles are imposed.

## 4. Results and Discussion

To validate the proposed methodology, three test scenarios were assessed:Experiment 1: Sample size. A synthetic dataset with 5 variables was utilized. The significant variability of probabilistic dependencies (0.005% being the rarest) theoretically requires 187,500 samples to achieve 95% confidence that the rarest event ($\bar{H},L,\bar{C}$, 1 in 62,500 events) will happen at least once.Experiment 2: Network complexity. A well-documented synthetic dataset with 12 variables and 12 edges was utilized, which makes possible to verify the behavior of the proposed method in DAG spaces that cannot be fully explored in reasonable time.Experiment 3: Real-world case. The proposed methodology was applied to electromyography (EMG) data during gait of 70 individuals, with and without patellofemoral pain syndrome (PFP). The cause of PFP is unknown, but indications of changes in EMG are known to exist.

All algorithms were implemented using Python 3.11 and executed on an Intel i7 12700F 12-core processor with 64GB of RAM.

### 4.1. Experiment 1: Sample Size

For this experiment, a BN consisting of 5 nodes and 5 edges was utilized, as illustrated in Figure 1. The BN was programmatically defined and used to generate a dataset containing 1,000,000 inferences, which served as the basis for subsequent experiments. For each analysis, a predefined number of randomly selected samples was used from this generated dataset.

A network comprising five binary variables needs at least $2^{5}$ samples to contain all possible relationships. However, the JPT shows that certain relationships are much more probable than others: p(L=1|H=0) only occurs once in 25,000 times and p(L=1|H=1) once in approximately 400 times. Therefore, to capture this relationship, more data are needed than p(B=1|H=1), which happens 1 in 20 times.

Initially, 100 samples were randomly selected from the original dataset. These 100 samples were then resampled 10 times, each resample consisting of 10 samples. A BN was learned from each resampled dataset, allowing us to observe the frequency of edges, as illustrated in Figure 2.

To verify the credibility of each edge’s existence according to the data, two techniques can be applied:Beta-Binomial approach: verify the probability that each edge a→b exists, being *a* and *b* dataset variables, and a≠b. Therefore, for 5 nodes, $(5^{2})-5=20$ distributions would be obtained.Dirichlet-Multinomial approach: verify the probability that a pair of variables are correlated in direction a→b, correlated in direction a←b, or are uncorrelated. Therefore, for 5 nodes, nCr(5,2)=10 distributions would be obtained.

#### 4.1.1. The Beta-Binomial Approach

In the Beta-Binomial approach, the causal direction of the edges is not considered, since the edges a→b and a←b are considered independent. In addition, it has a more significant number of distributions to be evaluated at the end, and an edge a→b would only be selected if it is present in more than 50% of the generated networks. As an example, we take the edge H→F, which is the most common, as seen in Figure 2, and evaluate the credibility of its existence using the Beta-Binomial approach. The posterior is shown in Figure 3.

The 95% credible interval of the existence of the edge H→F ranges from 14% to 29%, with mean 22%, although it is the most common edge in all the learned models. This leads to the conclusion that the Beta-Binomial approach fails to accurately model the distribution of the edges, since each potential direction of the edge is considered independently.

#### 4.1.2. The Dirichlet-Multinomial Approach

The Dirichlet-Multinomial method offers a more intuitive approach to determine the credibility of both the existence and direction of an edge. Rather than treating the distributions for each node pair independently, this method incorporates three factors: the existence of the edge and its direction (either right or left).

**100 samples**—The same data from the Binomial example were used: 100 resamples, 10 samples each, from 100 data points randomly drawn from the original dataset. The Posterior probabilities (from now on, instead of drawing posterior distributions, we will describe them in text. As an example, the Binomial distribution depicted in Figure 3 can be rewritten as H→F:22%(14%,29%)) are:
$$\begin{array}{cc} H\leftarrow F\;(10\;\mathrm{samples}):\; & 41\%(32\%,50\%);\\ H≁F\;(10\;\mathrm{samples}):\; & 38\%(29\%,47\%);\\ H\to F\;(10\;\mathrm{samples}):\; & 21\%(14\%,29\%). \end{array}$$

The Dirichlet-Multinomial approach seems much closer to reality than the Beta-Binomial one; the most probable occurrence is of H←F, however, by a small margin. All hypotheses have an overlapping credible interval, indicating that the learned model has a great deal of uncertainty. On the other hand, a more uncommon edge, such as H↔L, is seen to be non-existent:$$\begin{array}{cc} H\leftarrow L\;(100\;\mathrm{samples}):\; & 0.096\%(0\%,0.27\%);\\ H≁L\;(100\;\mathrm{samples}):\; & 96\%(93\%,99\%);\\ H\to L\;(100\;\mathrm{samples}):\; & 0.29\%(0.035\%,0.59\%). \end{array}$$

In this case, the hypothesis of non-existence of the relationship H≁L is considered certain, which is also expressed by the data. The credible interval of H≁L is much narrower and closer to 100%, but by any means represents the reality of the system that generated the data, caused by sampling bias. The overlapping credible intervals for the node pair (H,F) also indicate that there may be insufficiently or incorrectly sampled data. Increasing the number of samples is the only way to analyze the inner relationships between the variables of a system.

**1000 samples**—The proposed algorithm will now be evaluated with 1000 samples. This time, 1000 resampled datasets were generated, each containing 10,000 samples, maintaining the same proportion of resamples/datasets as in the previous example. The resulting Bayesian network is shown in Figure 4.

As anticipated, with the increased quantity of data points, the uncertainty in the distribution of edge existence decreases:$$\begin{array}{cc} H\leftarrow B\;(1000\;\mathrm{samples}):\; & 88\%(87\%,90\%);\\ H≁B\;(1000\;\mathrm{samples}):\; & 0.001\%(0\%,0.03\%);\\ H\to B\;(1000\;\mathrm{samples}):\; & 11\%(0.97\%,13\%). \end{array}$$

The 95% credible interval of H≁B is now nearly zero, suggesting that this edge likely exists (regardless of its direction); meanwhile, H←B and H→B have a mean of 88% and 11%, respectively, with a 95% credible interval range of less than 1%. This shows H←B as the edge most observed, although in the opposite direction compared to the reference model. Something similar occurs when observing the edge H↔L:$$\begin{array}{cc} H\leftarrow L\;(1000\;\mathrm{samples}):\; & 0.01\%(0\%,0.028\%);\\ H≁L\;(1000\;\mathrm{samples}):\; & 99.9\%(99.8\%,100\%);\\ H\to L\;(1000\;\mathrm{samples}):\; & 0.01\%(0\%,0.028\%). \end{array}$$

The probabilistic relation between (H,L) is very subtle. With this small amount of data, *H* is being seen as D-separated from *L* by B,F, and not as a common driver for both *B* and *L*. Such behavior is inherent in the data; the most common relations were discovered, with low levels of uncertainty, and can be seen as a good representation of the system. However, when observed by an expert, the low-probability edges could be selected as a subject of study in order to be discarded. Typical methodologies, such as Local Search or PC algorithm would only give a single ’optimum’ model, without any doubt on the real structure.

**1,000,000 samples**—As seen in the previous example, most of the edges present on the final network indicate an existing relationship present in the reference model, sometimes in the inverted direction. However, a specific relation (H←L) is much more rare. To verify whether increasing data provide a better model, we will now run the test with 1,000,000 samples, resampling 10,000 times in sets of 250,000 samples each. Figure 5 displays the maxima a posteriori for the appearance of each edge when utilizing 1,000,000 samples.

Utilizing the Dirichlet-Multinomial approach, we observe the adjustment in credible intervals with the addition of more data:$$\begin{array}{cc} H\leftarrow L\;(1000\;\mathrm{samples}):\; & 76\%(73\%,78\%);\\ H≁L\;(1000\;\mathrm{samples}):\; & 0.01\%(0\%,0.029\%);\\ H\to L\;(1000\;\mathrm{samples}):\; & 24\%(22\%,27\%). \end{array}$$

The maxima a posteriori is H←L, which is in the opposite direction compared to the reference model. However, it is still an improvement since, when using 1000 samples, this edge was set as non-existent. Also, the only edge with a MAP between 1% and 10% is of (B,L), which can be easily discarded by a specialist since this relationship is visibly caused by *L* being D-separated from *B* by *H* and *F*.

**hlSummary**—As the number of samples increased, it became clear how the credible interval changes and how it can be used to evaluate the uncertainty of the edges. Traditional methodologies often struggle to detect data-related issues, but modeling the direction of edges helped to uncover such problems.

### 4.2. Comparison with Alternative Methods

Using a Dirichlet-Multinomial model of edge existence and direction can provide a deeper understanding of the relationships on the complex system being observed. This metric can be extracted from both continuous and categorical cases, as long as a set of BNs learned from multiple subsets of the original data is available.

Other methodologies for extracting credibility do not offer the same kind of analysis. Some of these comparisons are discussed in Table 2.

### 4.3. Experiment 2: Network Complexity

LUCAS (*LUng CAncer Simple set*) (https://www.causality.inf.ethz.ch/data/LUCAS.html (accessed on 14 September 2024)) is a synthetic dataset generated by Bayesian networks with binary variables. The model shown in Figure 6 is generated by a Markov process; therefore, the state of each node is determined only by the parents. It has 12 variables: Smoking, Yellow_Fingers, Anxiety, Peer_Pressure, Genetics, Attention_Disorder, Born_an_Even_Day, Car_Accident, Fatigue, Allergy, Coughing, and Lung_Cancer. In particular, Born_an_Even_Day is completely independent of all others with probability 0.5 for each state, and Lung_Cancer is the target variable. This model does not have any biological logic, being completely synthetic.

An in-depth view of the most relevant edges produced by the proposed method is shown in Figure 7. Each line contains a pair of variables $\left(X_{1},X_{2}\right)$ from the LUCAS dataset and two probability distributions: the first indicates the existence of the edge in the model and the second indicates the direction. (This representation can be derived from the Dirichlet-Multinomial model as follows:**Edge existence probability**: summing the distributions for A→B and A←B, then calculating a binomial distribution against A≁B;**Edge direction probability**: assuming a binomial distribution when merging A→B and A→B).

These probability distributions are a significant advantage of the proposed methodology. Using them, we can extract the Maximum A Posteriori (MAP) estimate to generate the resulting graph, as shown in Figure 8. Notably, only three edges differ from the original model, and all of these variations occur within the Markov blanket of the *Smoking* variable. According to the FCI hypothesis, this discrepancy may suggest that the relationships among the variables *Smoking, Anxiety, Peer Pressure, and Yellow Fingers* might be mediated by an unknown fifth variable.

When compared to the Local Search and PC algorithms, the resulting networks differ. In practice, the color of the edges would not be known, as no prior information about the true structure is available. This is where probability distributions become valuable. For instance, the edge from Anxiety→Peer_Pressure appears in both networks derived from local search using two different scoring methods. However, when considering the probability of edge existence, this edge has a MAP below the 75% credibility threshold set for edge selection in the proposed methodology. Additionally, the PC algorithm, which relies on independence tests, failed to identify the relationship Genetics→Lung_Cancer, that our methodology indicates as highly probable.

#### Impact of Dataset Size on Inference Tasks

Since the LUCAS dataset is obtained from a predetermined JPD, the learning datasets were generated with $10^{3}$, $10^{4}$, $10^{5}$, and $10^{6}$ samples and evaluated against a testing dataset with $10^{3}$ samples by making an inference about the target variable Lung_Cancer. The results are displayed in Table 3.

The inference error in all datasets with more than $10^{4}$ samples is similar, with more than 80% correct outputs. However, with $10^{3}$ samples, only accurate outputs were obtained, indicating that the learned network incorrectly modeled the system.

### 4.4. Experiment 3: Real-Use Case

Instrumented gait analysis can provide comprehensive data on normal and pathological gait, which are useful in clinical practice to obtain information on joint motion, movements, timing, and action of the muscle, contributing to understanding the walking patterns and identifying the causes of gait irregularities [29,30]. The kinematic analysis of gait leads to a better perspective on how individuals use their combination of strength, flexibility, and muscle memory to achieve gait, allowing more direct approaches to diagnosing and treating any abnormalities [31]. On the other hand, knowledge of the activities of muscles during abnormal gait can help physicians support their diagnosis, design better surgical interventions, design and evaluate rehabilitation in a personalized manner, evaluate muscle fatigue, and support forensic medicine with objective results [30]. Despite the various clinical applications, instrumented gait analysis is still underutilized [30,31].

Many challenges still surround the comprehension of gait, mainly the EMG to kinematic coupling [32,33,34]. Combined with the lack of reference data and the difficulties in interpretability due to the large variability in the intrasubject [35,36], although there are a relevant number of studies [32,33,34,35,36] supporting the use of these signals in gait analysis, these factors limit its widespread use in routine clinical practice [29,30,31,37].

In recent years, different approaches have been pursued to overcome the aforementioned challenges. Machine Learning (ML) methods have been employed to analyze EMG signals, using different classifiers such as Support Vector Machines [38], Hidden Markov Models [39], Neural Networks [40,41] and others [42,43]. However, the adoption of these methods in the routine of healthcare professionals is still small [37,44].

The reasons for this narrow user base of ML models in the medical area are diverse, such as the strict regulations of medical protocols [37,44,45,46], the difficulty in updating models with new data or expert knowledge [37,44,45,46,47], and the lack of transparency of the models when making predictions [44,46,47]. This first topic is already being addressed by regulatory agencies—the FDA, for example, created guidance for AI systems [48]—but the last two are a sole responsibility of ML researchers.

Our methodology, as shown in previous examples, appears capable of overcoming such challenges, and aiming to evaluate it on a real-world case, we applied it on a dataset containing data from 70 runners during a 30 s session of running on a treadmill. Among the runners, 40 were healthy and 30 were diagnosed with patellofemoral pain (PFP). During each run, 8 myoelectric signals were collected: the electrical signals during the activations of the *biceps femoris* (BF), *gastrocnemius* (GASTRO), *gluteus maximus* (GMAX), *gluteus medius* (GMED), *rectus femoris* (RF), *tibialis anterior* (TA), *vastus medialis* (VM) and *vastus lateralis* (VL) muscles.

#### 4.4.1. Data Preprocessing

Surface EMG signals are formed by a series of high-frequency impulses obtained from the electric activity produced by skeletal muscles. An example is displayed on Figure 9. The EMG amplitude reflects the output of many spinal motor neurons [49], but understanding their relation is not simple. Therefore, the preprocessing of EMG signals is crucial.

Upon collection, the EMG signals were filtered with a zero-lag Butterworth filter. Then, the envelope was obtained using the Hilbert transformation. This transformation leaves any DC components untouched, and produces a 90 phase shift. Formally, the Hilbert transform *H* of a signal *u* can be expressed in terms of the Fourier transform *F* as [50]:(3)$$F\left(H\left(u\right)\right)\left(\omega \right)=\sigma_{H}\left(\omega \right)\cdot F\left(u\right)\left(\omega \right)$$
where,
(4)$$\sigma_{H}=\left\{\begin{array}{cc} i & \;\mathrm{for}\;\omega <0\\ 0 & \;\mathrm{for}\;\omega =0\\ -i & \;\mathrm{for}\;\omega >0 \end{array}\right.$$

Although the envelope has already smoothed the signal, it still has high-frequency components which will be removed with a zero-delay Savitzky–Golay (S-G) filter. The S-G filter is a digital moving-average filter, capable of smoothing the signal without distorting the original tendency [51]. Given a set of points $\left\{x_{j},y_{j}\right\}$, j=1,…,n, where $x_{j}$ is the independent variable and $y_{j}$ is the observed value, the S-G filtered signal $Y_{j}$ is [51]
(5)$$Y_{j}=\sum_{i=\frac{1-m}{2}}^{\frac{m-1}{2}}C_{i}y_{j+i},\frac{m-1}{2}\leq j\leq n-\frac{m-1}{2}$$
where $C_{i}$, i=1,…,m is the set of convolution coefficients which can be found in tables or calculated analytically [51]. For this work, m=7 was used.

Lastly, all EMG signals were resampled in order to decrease the sample-rate to 240 Hz using a polyphase FIR filter with coefficients calculated using a Kaiser window. A diagram containing all the steps for treating the EMG signals is shown in Figure 10.

#### 4.4.2. Per-Subject Model

Inter-subject variability is a characteristic of biological signals: although data collection used a strict methodology, the data from all individual subjects cannot be combined in a single model. The usual methodology is to learn a model for each subject, compare the models, and, using a specialist, combine them manually. This is presented in Figure 11.

Performing visual analysis, the relationship between VL and VM is consistent in these four BNs, although the direction on Subject 4 (blue) is the opposite of the others. Two other dependencies appear on the four BNs in Figure 11a: GASTRO ↔ TA and VL ↔ GMED. Another common characteristic in all of these networks is the existence of cluster nodes, i.e., nodes are parents of many others, such as VL, VM, and GASTRO. However, nodes such as TA, BF, RF, GMED, and GMAX have a maximum of one descendant.

BNs were then learned for EMG data collected from subjects with PFP, being four of the networks shown in Figure 11b. By inspecting the dependencies in the graphs and comparing them with healthy ones, it is expected to detect differences and similarities between muscle synergies in PFP and healthy individuals. Similarly to BNs in healthy individuals, the VM node acts as a cluster, having at least three dependencies on all of the networks. TA, RF, BF, and GMED also show little interaction with other nodes. However, in Subject 64 (magenta), GMAX acts as a cluster, which cannot be seen in any of the other BNs.

#### 4.4.3. Combined Model

The preceding analysis considered a subset of 4 out of the total of 40 subjects, due to the inherent complexity and difficulty associated with visualizing and comparing the entire network set. Our approach serves as a reliable and user-friendly technique for the quantitative analysis of combined Bayesian networks. To integrate all the Bayesian networks learned for healthy subjects, the credibility of each edge’s occurrence is illustrated in Figure 12.

The credible interval analysis indicates two major issues:**there is no relevant connection present in the data**: the *No edge* case is always the most likely when identifying the maxima a posteriori;**the dataset is too small to uncover any relevant connections**: the credible intervals are broad and exhibit significant overlap.

This discovery would not be achievable through mere analysis of the most prevalent edges; moreover, a quantifiable metric now exists for assessing the learned model’s reliability. The same problem arises when using data from pathological individuals, suggesting that the existing dataset is insufficiently large to detect probabilistic dependencies between muscle activations.

### 4.5. Challenges and Opportunities in Real-World Scenarios

A main challenge in applying data-driven BN learning to real-world scenarios is data variability. Fields such as biology, finance, energy, and climate studies often present significant variations between datasets, making it difficult to merge data from multiple sources into a unified model, and a single dataset is usually not enough to encode all possible relationships. For example, in electromyography (EMG) data, as discussed earlier, muscle activation patterns differ between individuals, even when data collection follows strict protocols. This variability requires learning separate BNs for each subject, manually comparing them, and integrating these models through expert analysis. Furthermore, limited datasets pose another significant challenge.

Incorporating domain knowledge into the BN learning process can help reduce the effects of small sample sizes and data variability. Credible interval analysis on the learned structure offers expert insight into model issues, highlighting where intervals are broad or overlapped. This provides a clearer indication that the dataset may be too small to reveal meaningful connections between specific variables, rather than resulting in empty or totally connected models, as is often the case with methods like Local Search, the PC algorithm, or analytical threshold techniques.

## 5. Conclusions

This paper presents a novel approach for learning Bayesian network structures using Markov chain Monte Carlo methods. The proposed method aims to address the challenge of model selection in the context of real-world data by providing a fully Bayesian approach that considers the credibility of each edge’s existence and direction in learned BNs. The algorithm employs a model-averaging strategy using bootstrapping to learn multiple distinct BNs from the data. An MCMC sampler is then used to define a Dirichlet-Multinomial model for each edge in the network, allowing for estimating the probability distribution of the edge’s existence and direction. Finally, a maximum a posteriori estimate is used to identify the most significant edges and their corresponding directions, which can be extracted by specialists to form a final DAG.

The results on synthetic datasets show that the proposed method achieves results similar to those of other methods while providing valuable information on the credibility of the learned model. By incorporating the inherent uncertainty in the learning process, the algorithm provides a range of plausible structures that can capture different aspects of the observed system. This approach acknowledges the complexity and elusive nature of a single accurate representation of the underlying relations in the system being modeled.

The proposed methodology demonstrated its utility in real-world scenarios. When applying the standard approach to learn a Bayesian network from electromyography data, there were no obvious indications that the data might be insufficient and that they could lead to incorrect conclusions. However, when evaluating the credible interval of the learned model, it became apparent that none of the proposed connections were considered credible. This example has shown how this approach in real-world scenarios solves existing issues, such as indicating if the dataset size is sufficient to ensure reliable results.

Identifying the direction of causal relationships proved to be a very challenging task, even in controlled synthetic scenarios. Nonetheless, the credibility metrics presented by the proposed method facilitate expert analysis, making it applicable to real-world scenarios in areas such as medicine, finance, and engineering, where data limitations are common and interpretability is crucial.

In general, the presented method offers a new way of learning BN structures from data, with potential applications in various domains where it is necessary to fully understand the statistical dependencies between variables. In future work, it would be beneficial to refine the algorithm and explore its performance on larger datasets, especially when applied to real-world use cases.

## Figures and Tables

**Figure 1 entropy-26-00829-f001:**
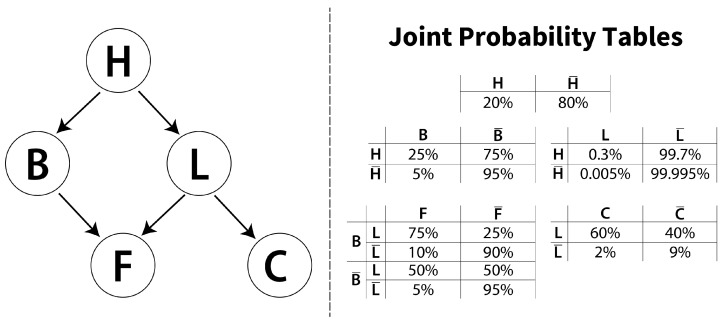
Example Bayesian network, adapted from Neapolitan [16]. This example has joint probabilities ranging from 75% to 0.005%, which requires a large dataset for containing all the possible relationships, making it ideal for evaluating the impact of dataset size on the confidence of learning a Bayesian network.

**Figure 2 entropy-26-00829-f002:**
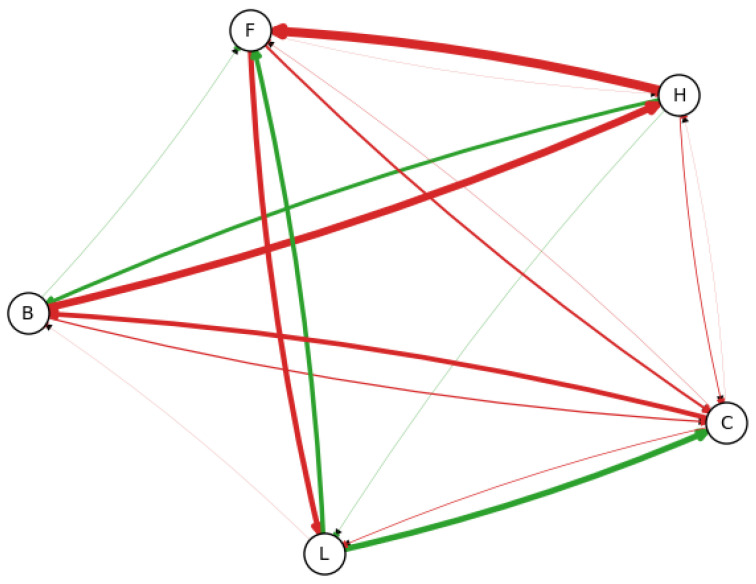
Observed frequency of each edge using 100 samples (100 resamples, 10 samples each), according to the edge width. As expected, relationships such as L→C and L→F are more frequent than H→L. The thickness of the edges indicate that there is not a convergence, and selecting the most probable edges would bring an almost fully connected graph. Edges colored in green are present in the original model, and edges colored in red are not.

**Figure 3 entropy-26-00829-f003:**
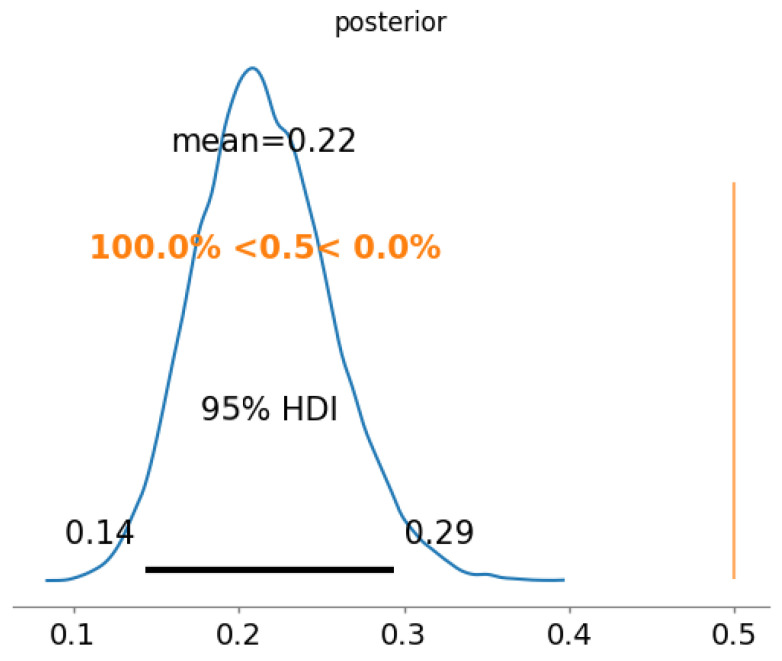
The posterior probability of the presence of the edge H→F when using 10 samples to learn the Bayesian network (BN). This particular edge is considered to have a low likelihood of existence, with a mean of 22% and a 95% credible interval ranging from 14% to 29%. The 95% credible interval, highlighted by the bold line at the bottom, remains entirely to the left of the 50% threshold indicated by the vertical orange line. Consequently, when evaluated by an estimator, it will be classified as non-existent.

**Figure 4 entropy-26-00829-f004:**
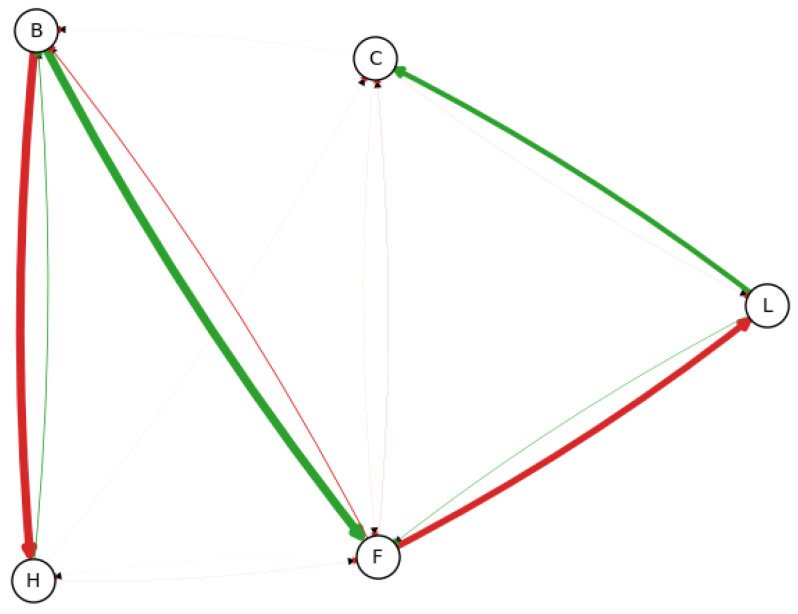
Observed frequency of each edge using 1000 samples, according to the edge width. Now, the edge concentration is much more precise, since now it only has a few thick edges. Edges colored in green are present in the original model, and edges colored in red are not.

**Figure 5 entropy-26-00829-f005:**
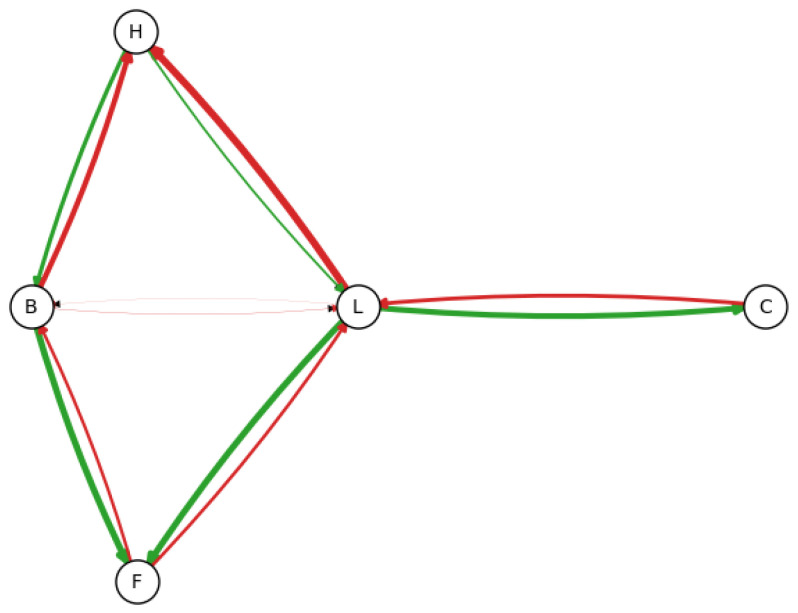
Observed frequency of each edge using 1,000,000 samples, according to the edge width. Looking at the frequency of the edges, we can see that some edges are more prevalent, including previously unseen (H←L). Green edges are present in the original model, and red edges are not.

**Figure 6 entropy-26-00829-f006:**
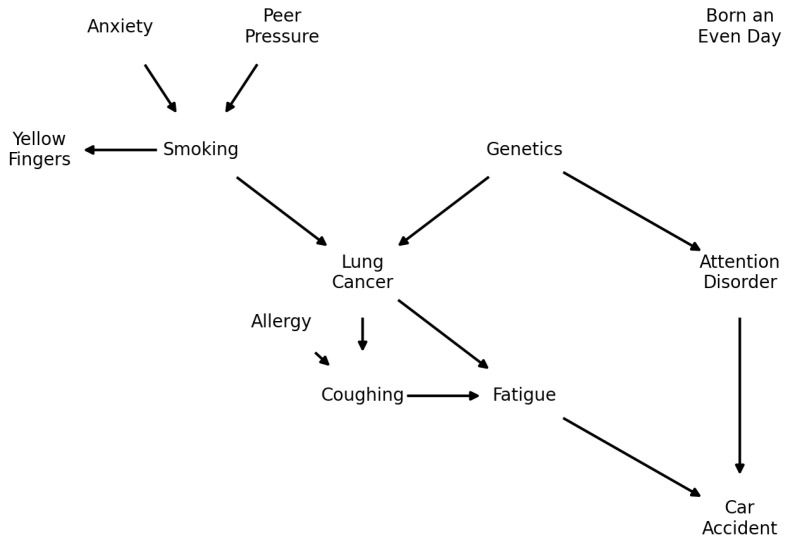
Graphical representation of the LUCAS network. It contains 12 variables, where *lung cancer* is usually treated as the target. It contains five colliders, two common-cause relationships and an independent variable, making it a useful benchmark for Bayesian networks structure learning algorithms.

**Figure 7 entropy-26-00829-f007:**
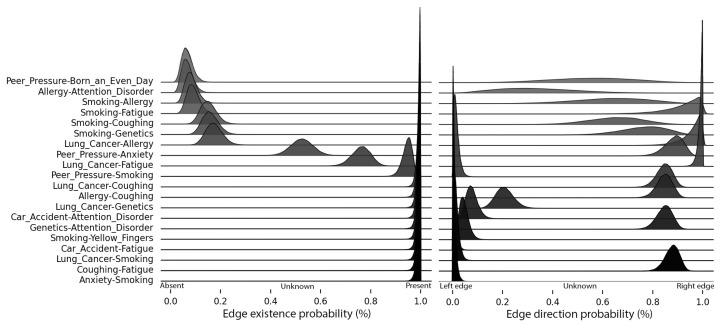
Probability distributions of edges existence (**left**) and direction (**right**) for the Lucas0 synthetic dataset. Edges with a low probability of existence can be seen to have a larger credible interval in evaluating their directions. The top edges are most likely absent, and the lower edges are most probably present in the evaluated model.

**Figure 8 entropy-26-00829-f008:**
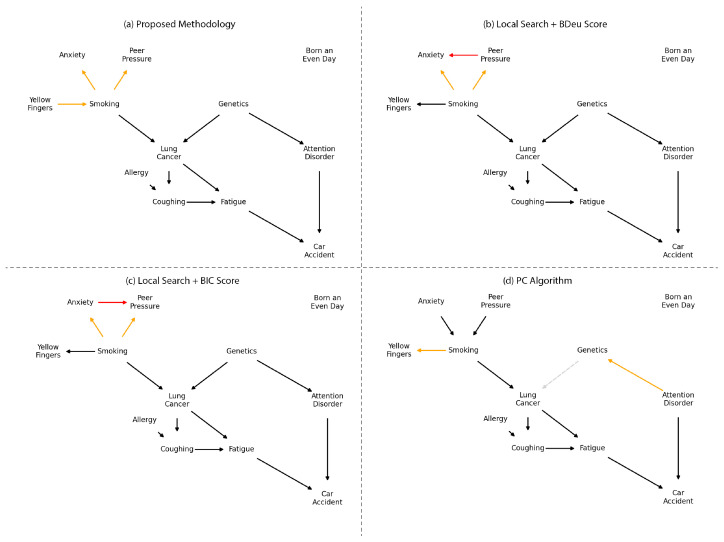
Method performance comparison with 10,000 samples, containing network structures learned using (**a**) the MAP of the proposed algorithm; (**b**) Local Search using the BDeu Score; (**c**) Local Search using the BIC score; (**d**) the PC algorithm. Black edges were found in the correct direction, orange edges were found on the opposite direction, dotted grey edges were not present, and red edges are spurious.

**Figure 9 entropy-26-00829-f009:**
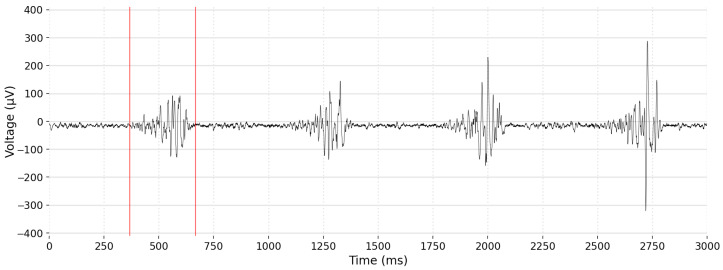
Raw EMG signal collected over the *vastus medialis* muscle, from Subject 9 with PFP, during the first 3 s of the session. The first major activation is delimited by the red markers. During activation, it contains spikes with negative and positive voltages, mirrored over the x-axis.

**Figure 10 entropy-26-00829-f010:**
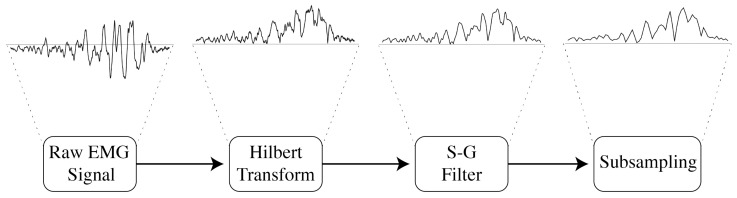
Diagram showing the steps for processing the EMG signals and the result of each step using as example the first major activation of the *vastus medialis* muscle, from Individual 9 with PFP.

**Figure 11 entropy-26-00829-f011:**
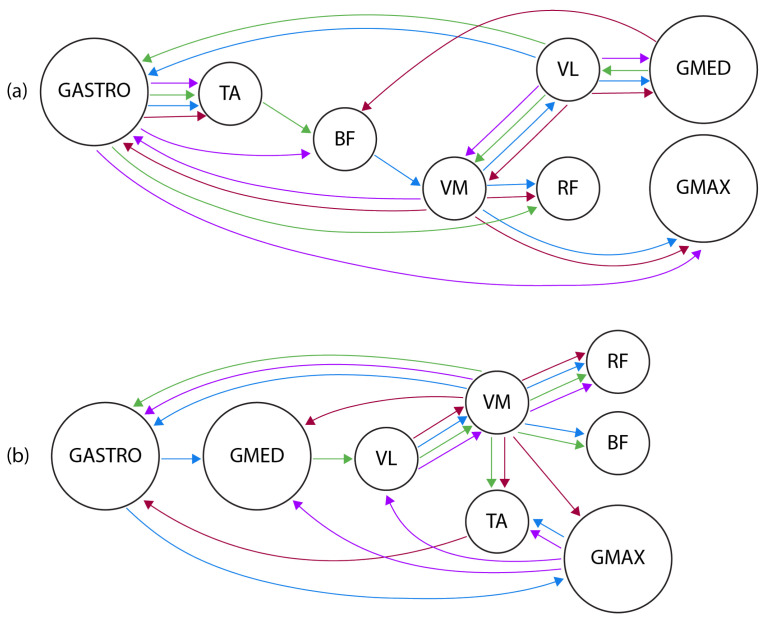
Bayesian networks of EMG relationships in healthy (**a**) and subjects with PFP (**b**). Color labels: (**a**) Red: Subject 1; Blue: Subject 4; Green: Subject 10; Magenta: Subject 40. (**b**) Red: Subject 49; Blue: Subject 53; Green: Subject 61; Magenta: Subject 64.

**Figure 12 entropy-26-00829-f012:**
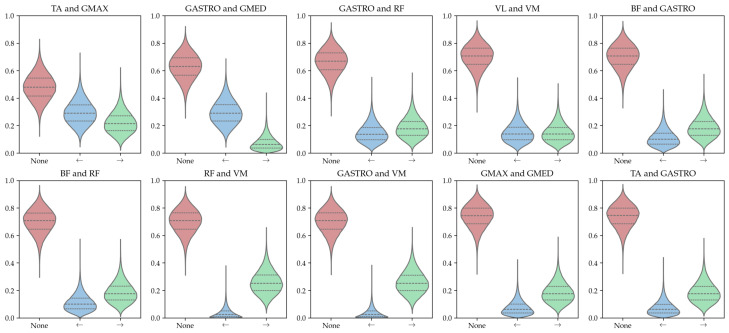
Posterior distribution plots of the 10 most credible edges among healthy runners, with the dotted lines indicate the 95% credible interval. Red shows the probability distribution of edge non-existence, blue shows existence in the left direction (e.g., *gluteus maximus* →*tibialis anterior* for the first plot), and green in the right direction (e.g., *tibialis anterior*→ *gluteus maximus* for the first plot). None of them have a maxima a posteriori which indicates their existence.

**Table 1 entropy-26-00829-t001:** Limitations of data-driven methodologies for BN structure learning in small sample size scenarios.

Algorithm	Limitations When Applied Small Sample Sizes
Score-based methods	Tend to overfit when data are scarce because they search for the highest-scoring DAG, potentially capturing spurious relationships.
Constraint-based methods	Independence tests used in PC and FCI algorithms can be unreliable with small sample sizes, leading to incorrect structures.
Threshold-based approaches	The threshold for selecting an edge will not be achievable, leading to structures with a small number of edges.

**Table 2 entropy-26-00829-t002:** Comparison of the proposed methodology with alternative metrics for assessing the quality of Bayesian network models.

Metric	Comparison
Analytic threshold	The threshold does not take into account uncertainty; Does not show any information on the credible interval of the relationships between variables.
Bayesian credible interval	Can only be used in continuous models; Manages the model level credibility by cutting out possible extreme events; Does not present insights on the quality of the model learned, only on the inference; Does not offer any information on the quality of the data used.
Information scores (BDeu and BIC)	Does not quantify uncertainty in predictions; Does not offer any information on the quality of the data used; Different structures can have the same score.
K-fold cross validation	Does not quantify uncertainty in predictions; Only evaluates the model as a whole, not offering information on the relationship between variables; Does not offer any information on the quality of the data used.

**Table 3 entropy-26-00829-t003:** Inference error between structures learned with the proposed algorithm. TP = True Positives; TN = True Negatives; FP = False Positives (Type-I Error); FN = False Negatives (Type-II Error).

Num. of Samples	TP	TN	FP	FN
$10^{6}$	65%	21%	7%	7%
$10^{5}$	65%	21%	7%	7%
$10^{4}$	68%	17%	10%	5%
$10^{3}$	72%	0%	28%	0%

## Data Availability

No data was generated by this study. Data used for the evaluation of the proposed algorithm are not available due to ethical restrictions.

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
