# Peer review of "Assessing Credibility in Bayesian Networks Structure Learning"

_entropy, 2024, doi:10.3390/e26100829_

Round 1

Reviewer 1 Report

Comments and Suggestions for Authors

This excellent paper explores in detail the use of a MCMC approach to constructing BN structures that have a measurable validity. The authors then apply this method to case data and discuss the potential benefits. While clearly the diversity of BN cases is yet to be explored this approach has promise to assist with advanced learning of complex systems especially in situations where the network structure is not understood by experts. 

My comments are very minor. I was surprised not to see earlier references to the work on Metrics for evaluating performance and uncertainty of Bayesian network models BG Marcot Ecological modelling 230, 50-62.  I would be keen to see synthetic cases generated from a known causal relationship and then tested with this approach. 

Author Response

Dear Reviewer,

Thanks for all of your comments! They were incredibly helpful, and we agreed with them all.  Attached, we're sending a detailed information on what was done to address them.

Best Regards.

Reviewer 2 Report

Comments and Suggestions for Authors

1It is recommended to more explicitly articulate in the introduction section the urgency and significance of the current research question, as well as the rationale for requiring a novel approach to assess the credibility of Bayesian Network (BN) structures. This would facilitate readers' quicker comprehension of the research background and its implications.

2While the article mentions some related work, a more systematic review and comparison of existing BN structure learning methods could be conducted, particularly highlighting their limitations in handling small sample data or complex network structures.

3When introducing the new method, its advantages over existing approaches should be clearly articulated, not just theoretically but also in terms of practical performance.

4Detailed descriptions of the experimental data sources, preprocessing procedures, and specific steps and parameter settings of the experiments are essential to verify the reliability and reproducibility of the experimental results.

5In addition to graphical presentations, it is advised to include textual summaries that generalize the key findings of the experiments, particularly the interpretation and discussion of the experimental outcomes.

6Comparative analyses of the new method against other existing methods (e.g., PC algorithm, local search algorithms) should be conducted to demonstrate in which aspects the new method outperforms and why.

7Discuss the limitations of the current research methodology, such as potential biases and dependence on data quality.

8Address the potential and challenges of applying the new method in real-world scenarios, as well as strategies for integrating it with domain knowledge to address practical problems.

Author Response

(The authors gave the same response as above.)

Round 2

Reviewer 2 Report

Comments and Suggestions for Authors

The author has answered the relevant questions, and the paper is carefully revised and recommended for acceptance.